# Safety, Feasibility and Technical Considerations from a Prospective, Observational Study—CIREL: Irinotecan-TACE for CRLM in 152 Patients

**DOI:** 10.3390/jcm11206178

**Published:** 2022-10-19

**Authors:** Thomas Helmberger, Pierleone Lucatelli, Philippe L. Pereira, Aleksandar Gjoreski, Ivona Jovanoska, Zoltan Bansaghi, Stavros Spiliopoulos, Francesca Carchesio, Dirk Arnold, Andreas Baierl, Bleranda Zeka, Nathalie C. Kaufmann, Julien Taieb, Roberto Iezzi

**Affiliations:** 1Institut für Radiologie, Neuroradiologie und Minimal-Invasive Therapie, München Klinik Bogenhausen, Englschalkinger Str. 77, 81925 München, Germany; 2Vascular and Interventional Radiology Unit, Department of Radiological Oncological and Anatomopathological Sciences, Sapienza University of Rome, 00161 Roma, Italy; 3SLK-Kliniken Heilbronn GmbH, Zentrum für Radiologie, Minimal-invasive Therapien und Nuklearmedizin, Am Gesundbrunnen 20-26, 74078 Heilbronn, Germany; 4Diagnostic and Interventional Radiology, General City Hospital “8th September”, 1000 Skopje, North Macedonia; 5Medical Imaging Center, Semmelweis University, 1082 Budapest, Hungary; 6Interventional Radiology Unit, 2nd Department of Radiology, School of Medicine, National and Kapodistrian University of Athens, Rimini 1, 124 62 Athens, Greece; 7Dipartimento di Diagnostica per Immagini, Radioterapia Oncologica ed Ematologia, UOC di Radiologia Diagnostica ed Interventistica Generale, Fondazione Policlinico Universitario “A. Gemelli” IRCCS, 00168 Rome, Italy; 8Asklepios Tumorzentrum Hamburg, AK Altona, 22763 Hamburg, Germany; 9Department of Statistics and Operations Research, University of Vienna, 1010 Vienna, Austria; 10Clinical Research Department, Cardiovascular and Interventional Radiological Society of Europe, Neutorgasse 9, 1010 Vienna, Austria; 11Assistance Publique Hôpitaux de Paris, Service d’Hepatogastroentérologie et d’Oncologie Digestive, Hôpital Européen Georges Pompidou, Université Paris Descartes, Sorbonne Paris-Cité, 75015 Paris, France; 12Istituto di Radiodiagnostica, Università Cattolica del Sacro Cuore, 00168 Rome, Italy

**Keywords:** CRLM, mCRC, TACE, DEBIRI, irinotecan

## Abstract

CIREL, a prospective, Europe-wide, observational study aimed to assess the real-world feasibility and tolerability of irinotecan-based transarterial chemoembolization (LP-irinotecan TACE) for unresectable colorectal cancer liver metastases with regard to the treatment plan and adverse events (AEs). CIREL enrolled 152 eligible patients (≥18 years) with liver-only or dominant metastases treated with LP-irinotecan TACE following a multidisciplinary tumor board decision. Data were prospectively collected for baseline, the number of planned and performed sessions, and technical information and safety according to CTCAE 4.03/5.0. Results from 351 analyzed treatment sessions showed technical success for 99% of sessions, and 121 patients (79%) completed all planned sessions. Further, 60% of sessions were performed using opioids, 4% intra-arterial anesthetics, and 25% both. Additionally, 60% of patients experienced at least one peri-interventional AE of any grade; 8% of grade 3–4. Occurrence of AEs was related to larger liver-involvement (*p* < 0.001), bi-lobar disease (*p* = 0.002), and larger beads (*p* < 0.001). Using corticosteroids together with antiemetics showed reduced and lower grade vomiting (*p* = 0.01). LP-irinotecan TACE was tolerated well and had a high proportion of completed treatment plans. This minimally invasive locoregional treatment can be used together with concomitant systemic therapy or ablation.

## 1. Introduction

A majority, 80%, of patients with metastatic colorectal cancer (mCRC) confined to liver metastases (CRLM) are not eligible for surgery or local thermal ablation. In these patients, disease control via anti-cancer medical treatment represents the main treatment goal [1]. Combination chemotherapy regimens, including fluoropyrimidines with irinotecan and/or oxaliplatin and monoclonal antibodies, are recommended as first and second line treatments in most cases with good clinical condition. Depending on the patient’s molecular markers, anti-EGFR for RAS and BRAF wildtype patients and bevacizumab, an anti-angiogenic drug, for RAS mutant patients is generally added to chemotherapy. New treatment options have emerged for some molecular subgroups, e.g., patients with tumors harboring BRAF V600E mutations, microsatellite instability status, and others [2,3].

However, for most patients, those strategies result in either a continuous administration of systemic treatment (although “induction and maintenance”-strategies have gained importance [4,5]), or progressive disease after second line creating a situation without attractive further treatment options for the vast majority of patients [6,7]. 

For patients with liver-only or liver-dominant lesions, irinotecan-based transarterial chemoembolization (irinotecan-TACE) has to be considered as part of the toolbox of loco-regional treatment strategies [1]. It can be used together with systemic treatment or alone [8,9,10] and may be used in first or later line therapy. Irinotecan-TACE is generally well tolerated with serious adverse events (AEs) experienced in about 0–10% of patients [11,12,13,14,15,16,17]. However, grade 1 and 2 post-embolization syndrome (PES) after a treatment session is a common occurrence (10–60%) [17,18,19,20,21,22,23,24]. Although there are recommendations regarding treatment plans [25,26], such as to perform at least two sessions for unilobar and four sessions for bilobar disease and regarding technical considerations like the use of opioids, intra-arterial anesthetic, and antiemetics, recent literature has highlighted the lack of standardized technical protocols [11]. This likely contributes to the wide range of PES AE rates reported in the literature. Additionally, non-standardized protocols and patient selection criteria across studies make comparisons difficult and therefore limit the easy application of published information to the real-life practice. On top of that, it is important to improve our understanding of how irinotecan-TACE is used in real-life, as well as identify factors that make patients more vulnerable to PES and the most the suitable procedural management to further increase the tolerability and minimize treatment-associated burden for the patients.

The prospective, observational study CIrse REgistry for LifePearl microspheres (CIREL) was designed and conducted by multidisciplinary teams, including interventional radiologist and oncologists, and captured real-life data of 152 CRLM patients treated with irinotecan-TACE using irinotecan-eluting microspheres (LP-irinotecan TACE). This analysis was performed as a part of the CIREL study protocol in order to provide a full-cohort summary of the use at different stages of the CRLM cancer continuum care and evaluate the feasibility and safety of the procedure, in terms of completed treatment plans and occurrences of AEs. Further, the aim of our analysis was to identify the potential association between more frequent or more severe AEs and characteristics, such as disease severity or treatment characteristics.

## 2. Materials and Methods

### 2.1. Study Design/Setting

The CIREL rationale and methodology was published by Pereira et al., 2020 [27]. Hence, 152 patients were prospectively and consecutively enrolled by 20 centers in 11 European countries. All treatments were performed by interventional radiologists who had experience with the treatment and had performed at least 40 treatments in their center or 10 treatments with any drug-eluting beads in the 12 months prior to center enrolment. 

### 2.2. Patients

Eligible patients were  ≥18 years with histologically confirmed colorectal adenocarcinoma with liver-only or liver-dominant metastases, and scheduled for treatment with LP-irinotecan TACE (LifePearl microspheres, Terumo Europe N.V., Leuven, Belgium) as decided in a multi-disciplinary tumor (MDT) board. Patients were enrolled between February 2018 and August 2020. 

Technical aspects regarding the treatments were performed at the treating physician’s discretion and patients may have received prior intra-arterial treatment.

### 2.3. Study Objectives and Data Sources

The primary objective was analyzing the real-life use of LP-irinotecan TACE via the treatment intention defined as follows: first line or consolidation treatment after response to first line systemic therapy,use of LP-irinotecan TACE in combination with ablation in a curative intent,as an intensification of treatment with/without concomitant therapy for chemo-refractory patients still eligible for further systemic treatment,as salvage treatment in progressive patients for chemo-refractory patients non-eligible for further systemic treatment.

Patient were defined as non-eligible for further treatment, i.e., salvage therapy patients, in case of progressive disease after treatment with irinotecan, oxaliplatin, anti-angiogenics and anti-EGFR-antibodies (in case of RAS and BRAF wt) or were defined as salvage therapy patients by the treating physicians. In contrast to a previously published methodology [27], observed treatment intention groups were slightly restructured to create meaningful sample sizes and align with the most recent guidelines. 

As a secondary objective feasibility, procedural data and safety were analyzed. Feasibility was evaluated in terms of analysis of treatment plans. In detail, the number of procedures planned to be performed, as well as whether unilobar (treatment sessions only in the right or only in the left liver lobe) or bilobar (treatment sessions in both liver lobes) was planned, were collected and analyzed with respect to the actually performed number of treatment sessions and actually performed unilobar or bilobar treatment of the patients.

Procedural data—regarding bead size, the total dose infused –, peri-procedural management, as well as technical success (defined as complete delivery of the planned dose, complete stasis or both) were collected and analyzed. 

For the analysis of safety, the occurrence of AEs was registered and evaluated. AEs were classified according to CTCAE 4.03 and 5.0 (Cancer Institute’s Common Terminology Criteria for Adverse Events) and were collected continuously. AEs reported on the day as any LP-irinotecan treatment session were defined as peri-interventional and AEs reported within 2–30 days of any treatment session or reported on the same day as any LP-irinotecan treatment session but lasted longer than 24 h were defined as acute AEs. Pain, nausea, vomiting, and fever were counted as PES when they occurred on the same day as any LP-irinotecan treatment session.

Biologic AEs were defined using laboratory values that were collected periodically during the treatment cycle (i.e., within 30 days of a treatment session) and were analyzed when the CTCAE grade increased compared to baseline. For this, abnormal laboratory values were identified using the reference ranges in Appendix A. All biologic AEs after completion of the treatment plan were collected using the AE electronic case report form (e-CRF). 

Clinical and technical data were collected via e-CRFs using OpenClinica 3. Automated data checks and verifications were employed via a separate data management system and remote monitoring was conducted quarterly to discuss internal data logic.

### 2.4. Statistical Methods

To describe and compare safety and toxicity, the proportion of patients or sessions with at least 1 AE (≥1 AE), as well as the AE burden score (AE-BS) were used. The AE-BS was employed to analyze the cumulative burden to a patient and was calculated according to Le-Rademacher et al., 2020 [28]. Briefly, the AE-BS represents the weighted sum of all AEs and the respective weighted grades. Since all AEs were graded according to CTCAE, which reflects comparable severity between AE types, the weight of an AE type was defined as the CTCAE grade. Therefore, the AE-BS was calculated by taking the sum of all AE grades per patient. Categorical data is presented as counts and percentages. For continuous data, median (range) is shown. Significant differences between categorical data were assessed using the Chi-squared test and the Kruskal–Wallis Test (*p* values ≤ 0.05 were considered significant). A multivariable analysis was performed using a binary logistic regression model whereby the predictor variables were determined following the univariable Chi-squared test and a stepwise variable selection procedure with a significance level of 0.1 for entering the predictor into the stepwise model. Additionally, the total number of sessions performed was added to adjust for the connection between bilobar/unilobar treatment and the total number of sessions performed.

Data were analyzed and plotted using RStudio version 1.2.1335 under R4.1.1. The Chi-squared test, Kruskal-Wallis test and the multivariable binary logistic regression were performed using IBM SPSS Statistics 25.

## 3. Results

### 3.1. Baseline Characteristics and Patient Selection

Briefly, 152 CRLM patients with a median age of 66 years (range 37–90) were enrolled, of which 93 (61%) were male and 66 (43%) also had extra-hepatic disease. Further, 71 (47%) had 1–3, 52 (34%) had 4–10, and 29 (19%) had >10 liver lesions. Liver involvement was <25% in 82 (54%), 25–50% in 59 (39%), and >50% in 11 (7%) patients. Moreover, 87 patients (57%) had liver lesions located in both liver lobes, 54 (36%) only in the right lobe, and 11 (7%) only in the left lobe. Baseline characteristics are summarized in Table 1.

LP-irinotecan TACE was most frequently used in patients that had not responded to prior systemic treatment: in 41 (27%) as “intensification treatment with/without concomitant therapy” for patients that were still eligible for further systemic treatment, and in 46 (30%), and as “salvage therapy in progressive patients” for chemo-refractory patients that were non-eligible for further systemic treatment. However, for 41 (27%) and 19 (13%) patients, LP-irinotecan TACE was either used as “first line treatment or consolidation treatment” after response to first line systemic therapy or as “combination treatment with ablation with curative intent”, respectively.

### 3.2. Treatment Feasibility and Technical Success

Briefly, 73 (48%) patients were planned to receive unilobar treatment, most frequently with one or two sessions (24 and 44 patients, respectively), whereas 3 sessions were planned for two patients and four sessions for three patients. Further, 79 (52%) patients were planned to receive bilobar treatment, most frequently with two or four sessions (18 and 55 patients, respectively) and only one and five patients were planned to receive one and three sessions, correspondingly (Figure 1). Additionally, 121 patients (80%) completed all planned treatment sessions while 31 (20%) patients did not. For 13 of those, the treatments were aborted due to progression or bad condition of the patient, nine patients either refused further treatments or could not be further contacted by the centres, for six patients the reasons were not available, and three patients withdrew consent for data collection therefore it could not be verified whether the treatment sessions had been completed as planned. The vast majority of patients (28; 90%) aborting the treatment plan were supposed to receive a bilobar treatment and 16 (10%) of those received only unilobar treatment as a result. Four treatment sessions were planned for 55 patients with bilobar treatment but only 34 patients received four sessions. For unilobar treatments, we saw no meaningful difference between the feasibility of the most common plans (one or two sessions). Ultimately, this resulted in 351 analyzed treatment sessions with bilobar treatment for 63 patients and unilobar treatment for 89 patients.

For most treatments (*n* = 270; 77%), small beads (100 µm) were used and the most common loading dose was 100 mg (246 sessions; 70% Table 1, Figure 1). Further, 200 µm beads were used mostly for patients with >25% liver involvement (Figure 1). One treatment session was performed using 400 µm beads loaded with 100 mg irinotecan for unilobar treatment with 25–50% liver involvement (not included in Figure 1).

Technical success was reported for 346 (99%) treatment sessions as defined by either completely delivering the dose (212; 60%), complete stasis (59; 17%), or both (75; 21%).

### 3.3. Safety and Toxicity

Briefly, 91 (60%) patients experienced at least one AE occurring on the day of the treatment session (peri-interventional), and 49 (32%) patients at least one acute AE between 2 and 30 days after a treatment session (Table 2). The most common peri-interventional AEs experienced were related to PES with grades 1 and 2 pain, nausea/vomiting occurring after 121, and 51 sessions, respectively. Severe (grade 3) pain and nausea/vomiting were only reported after four and eight sessions, respectively. Biologic AEs were only grade one and two and mainly hepatobiliary disorders with increased alkaline phosphatase levels occurring after 32 sessions and increased ALT and AST levels occurring after 24 and 22 sessions, followed by occurrences of hypoalbuminemia after 28 session and decreased lymphocytes after 27 sessions.

The AE-BS as a cumulative burden of AEs experienced per patient after each session is summarized in Figure 2. For peri-interventional and acute/biologic AEs, 89% and 95% of sessions resulted in a score between 0 and 3, 9% and 3% between 4 and 6, and 1% and 2% above 6. For both, the median AE-BS was the highest after the first treatment session. The respective heatmaps show the AE-BS per treatment session with every row representing an individual patient which depicts that while some patients will experience AEs only after the second, third, or fourth sessions, most patients experience a constant AE-BS during the treatment cycle and others reported no AEs at all (AE-BS of 0).

### 3.4. Relationship between Baseline or Treatment Characteristics and AE Events

To identify which baseline- and treatment-specific factors could be related to a higher occurrence of AEs, the proportion of patients and treatment sessions with ≥1 AE was used (Appendix A). For peri-interventional AEs, higher percentage of liver involvement (*p* = 0.001), bilobar treatment (*p* = 0.01), larger bead sizes (*p* < 0.001), and receiving LP-irinotecan TACE as the first session (*p* = 0.01) were significantly associated with ≥1 AE. More treatment sessions per patients and higher doses of irinotecan were not found to increase peri-interventional AEs. Neither were prior intra-arterial treatment of the liver nor prior use of irinotecan as systemic chemotherapy. 

For acute AEs, higher ECOG score (*p* < 0.001) and bilobar treatment (*p* = 0.03) were significantly related to the presence of AEs.

Next, a multivariable binary logistic regression was conducted to further analyse the adjusted relationship between the aforementioned identified factors and treatment sessions with ≥1 AE. The results confirmed the association of bilobar treatment (odds ratio 3.1; *p* = 0.002), >25% liver involvement (odds ratio 2.9; *p* < 0.001), >100 µm bead size (odds ratio 3.9; *p* < 0.001) with an increase in sessions with ≥1 peri-interventional AEs. Neither increase in dose nor the number of performed sessions were associated with increased ≥1 peri-interventional AEs (Table 3).

For acute AEs, the multivariable logistic regression showed that ECOG > 0 (odds ratio 3.3; *p* < 0.001) was associated with more AEs and confirmed that dose and the total number of sessions performed were not associated with more AEs (Table 4).

### 3.5. Relationship between Procedural Medication and PES AE-BS

The types of procedural medications could be divided into 6 groups depending on what combinations were used to manage AEs related to PES (Figure 3). Pain was managed using opioids alone, opioids together with intra-arterial anesthetic or intra-arterial anesthetic alone. Against nausea/vomiting, either antiemetics or antiemetics together with corticosteroids were used. The procedures were performed using the following combinations: 125 (36%) only opioids, 85 (24%) opioids and antiemetics, 15 (4%) only intra-arterial anesthetic, 33 (9%) intra-arterial anesthesia together with opioids, and 55 (16%) opioids with intra-arterial anesthetic, corticosteroids, and antiemetics. Further, 38 (11%) could not be divided into groups with meaningful sessions numbers and are therefore listed as “Others”. 

Sessions where only intra-arterial anesthetic was used caused ≥1 PES AEs most often (10/15; 67%) and had a median PES AE-BS of 1. Sessions where intra-arterial anesthetic together with opioids was used had the lowest occurrence of ≥1 PES AEs (10/30; 30%) and had a median PES AE-BS of 0. Opioids alone, opioids together with antiemetics, or all four groups combined were associated with similar frequencies of ≥1 PES AEs, 49/125 (39%), 36/85 (42%), and 26/55 (47%) and median AE-BS of 0. While we could not find significant differences in the proportion of ≥1 PES AEs between the groups, there were significant differences between median PES AE-BS across all groups (*p* = 0.036). Taking into account the heterogeneous data set and observational nature of the data collection, no post-hoc pair-wise comparison was performed. However, the groups were further analyzed to assess bias due to association with the factors that were previously identified to be connected with ≥1 PES AEs (Appendix A). While sessions performed with intra-arterial anesthetic without opioids (group 3) were associated with most of those factors, such as the use of 200-µm beads, only the percentage of liver involvement of 25–50% could be exclusively associated with group 3 (intra-arterial anesthetic without opioids). 

Next, we compared the number of PES AEs reported after treatment sessions performed without corticosteroids or with corticosteroids and antiemetics (Table 5), finding significantly lower reports of vomiting (*p* = 0.01) and reduced reports of grade 1 pain (*p* = 0.01), but increased reports of grade 2 pain (*p* = 0.009) when corticosteroids were used.

## 4. Discussion

CIREL was designed to capture the real-life use of LP-irinotecan TACE for CRLM across Europe to improve the evidence-base and support the real-world practice of CRLM treatment. Historically, irinotecan-TACE was associated with salvage therapy for patients who had failed first, second, and third-line systemic chemotherapy. However, due to its favorable toxicity profile and high local efficacy, irinotecan-TACE can be an attractive loco-regional treatment for earlier stages of the continuum care for CRLM cancer, such as in combination with systemic therapy or to obtain a sustained disease control, allowing halt of treatment (e.g., “chemo-holidays”) and/or to alleviate the toxicity-burden of maintenance systemic treatment [29,30]. The interim analysis of the first 50 patients enrolled in CIREL has already reported the use of LP-irinotecan TACE beyond the former indications limited to “salvage” therapy settings, such as in combination with ablation, with curative intent, and even as a first-line treatment [31]. Results from the whole cohort presently confirm these observations with 70% of patients treated with intent different than salvage therapy. 

The favorable toxicity profile of drug eluting microspheres is supported by other studies reporting low incidence of serious AEs (0–10%) [15,16,21,22], which is now confirmed in a prospective study with our results of serious AEs (CTACE grade 3 or 4) in 8% on the same day and 14% within 30 days of treatment sessions. Biologic AEs were also indicative of mostly mild (grades 1 and 2) increases in factors associated with liver toxicity. Furthermore, we used the AE-BS to more accurately explore the burden caused to patients by the presence of multiple mild or severe AEs. Overall, the AE-BS per patient ranged from 0 to 3 in most cases, with only a few individuals having a score above 6, and no cumulative effect of multiple treatment sessions could be observed. Further, no systemic-treatment related AEs, such as neurotoxicity or peripheral neuropathy, were reported. 

Despite the low occurrences of grade 3 and 4 AEs, increasing the awareness of what factors are related to more frequent AEs, how to apply preventive methods using peri-procedural medications and the appropriate pre-procedural consultation, as well as post-procedural monitoring are important to ensure the completion of the treatment plan and maintain the patients’ quality of life. Our results show that 60% of patients had at least 1 peri-interventional AE, the vast majority PES-related. Indeed, most studies report PES as the most frequent AE. However, the occurrence rates vary between 10% and 60% [11] and some guidelines even classify PES as expected outcome and not as an AE [13]. This is also illustrated when we compare our full-cohort results to the interim results from the first 50 patients enrolled to CIREL as the proportion of patients with mild AEs (grade 1 or 2) was 26% [31]. After CIREL placed special emphasis on the documentation of all mild and severe AEs, we were able to confirm that most mild peri-interventional AEs (grades 1 and 2) were related to temporary PES such as pain, nausea, vomiting or fever, and to analyze how baseline and treatment characteristics relate to those. Therefore, while our results correspond to the upper limit of the ranges reported in the literature, we believe that they can contribute to reducing the common occurrence of PES to greatly benefit patients. However, a major obstacle recognized by interventional radiologists performing irinotecan-TACE for CRLM patients is the overt lack of CRLM-focused standardized routines [25,32,33,34]. Technical details, such as the choice of bead size and loading dose, are crucial to ensure the complete delivery of the target dose, which together with the completion of the treatment plan is the foundation for achieving the best results. 

Moreover, the need to differentiate TACE protocols for hepatocellular carcinoma (HCC) from TACE protocols for CRLM has been highlighted [33,34]. First, ensuring the completion of the treatment plan can be challenging in patients with CRLM as they are believed to be more susceptible to PES than HCC patients. Despite recommendations for two sessions and four sessions for unilobar and bilobar treatment, respectively, our data show that in the real-world for unilobar treatments, 1 individual session and for bilobar treatment two or three individual sessions can be planned. Although there was a variability of how often sessions were performed, a high proportion of patients (80%) completed all planned sessions. However, 38% of patients with a bilobar treatment plan were not able to complete the treatment plan due to the decreasing state of the patients’ health or due to the patients’ refusal to further treatments. 

We found that patients with a higher percentage of liver involvement or with a disease extended to both lobes (and therefore were planned for bilobar treatment), also experienced peri-interventional AEs more frequently. Additionally, the first session was associated with significantly more occurrences of ≥1 peri-interventional AEs and the effect of decreased odds for the occurrence of AEs when more sessions were performed was further confirmed by our multivariable analysis. This could be partially attributed to the high number of patients with plans for bilobar treatment that aborted the treatment plan after the first session (Figure 1) and to the treating physicians adapting the medication doses of sequential sessions. 

Assessing the effectiveness of one or two sessions for uni- and bilobar treatments, respectively, will give crucial insight to help optimize treatment planning. With that information, the possibility to reduce the number of sessions or adapt to on-demand treatment planning could be explored more safely.

Another differentiation to HCC protocols should be the recommendations for bead sizes. According to the most recent evidence and since CRLM are less hypervascular compared to HCC, smaller bead sizes should be used as they have the potential to deliver the dose deeper into the tissue [35,36,37,38]. Studies found that smaller beads are not connected to more AEs [16,23,35,39]. In our cohort, small beads were used for 77% of procedures while larger beads were used for more advanced stages of the disease (bilobar disease and higher percentage of liver involvement). We also found that more sessions resulted in ≥1 peri-interventional AEs when larger bead sizes were used.

Regarding the association between AEs and dose, one study found a relationship between the injection of more than 100 mg irinotecan and increased toxicity [40]. However more recent studies could not verify the association between dose and AEs [12,41,42]. Our results do not indicate increased toxicity for higher doses but could show decreased odds of occurrence of AEs associated with higher doses. Of note, all sessions that used >100 mg irinotecan were performed by one center, and therefore due to potential bias no meaningful conclusions can be drawn about this effect.

One other major factor that should be considered when discussing AEs is the type of procedural medications used. We aimed to shed light on what procedural medications should be used to limit the most commonly experienced PES AEs and found that using intra-arterial anesthetic without opioids resulted in a higher median PES AE-BS score. Further, the use of corticosteroids together with antiemetics was connected to significantly lower incidence of vomiting but higher incidence of pain. Steroids are suggested by some but not all TACE recommendations for CRLM [25,26]. Since corticosteroids are routinely used in cancer therapy settings and have not been described to cause pain, there likely is no causal association between the higher incidence of pain and the use of corticosteroids.

### Limitations

While multi-center, non-interventional data provide a very suitable reflection of the real-world scenario, limitations due to potential bias resulting from uncontrolled confounding factors have to be acknowledged. Especially when trying to identify causal associations between AEs and patient- and treatment-related factors or procedural medications, we have to be apprehensive of potential bias introduced due to unequal groups, as well as center-specific under- or overreporting of AEs. 

## 5. Conclusions

Being feasible, safe, with a high technical success rate, and proposed beyond salvage indications, our analysis confirmed that LP-irinotecan TACE represents an adequate treatment option for patients with CRLM. However, it is used in different settings with different protocols in terms of procedural medications. The low occurrence of AEs likely contributes to the high proportion of completed treatment plans. The main factor for aborted treatment plans was bilobar disease. Understanding the correct procedural medications and identifying patients who are more vulnerable to AEs could allow us to offer an effective and safer procedure to patients with CRLM.

## Figures and Tables

**Figure 1 jcm-11-06178-f001:**
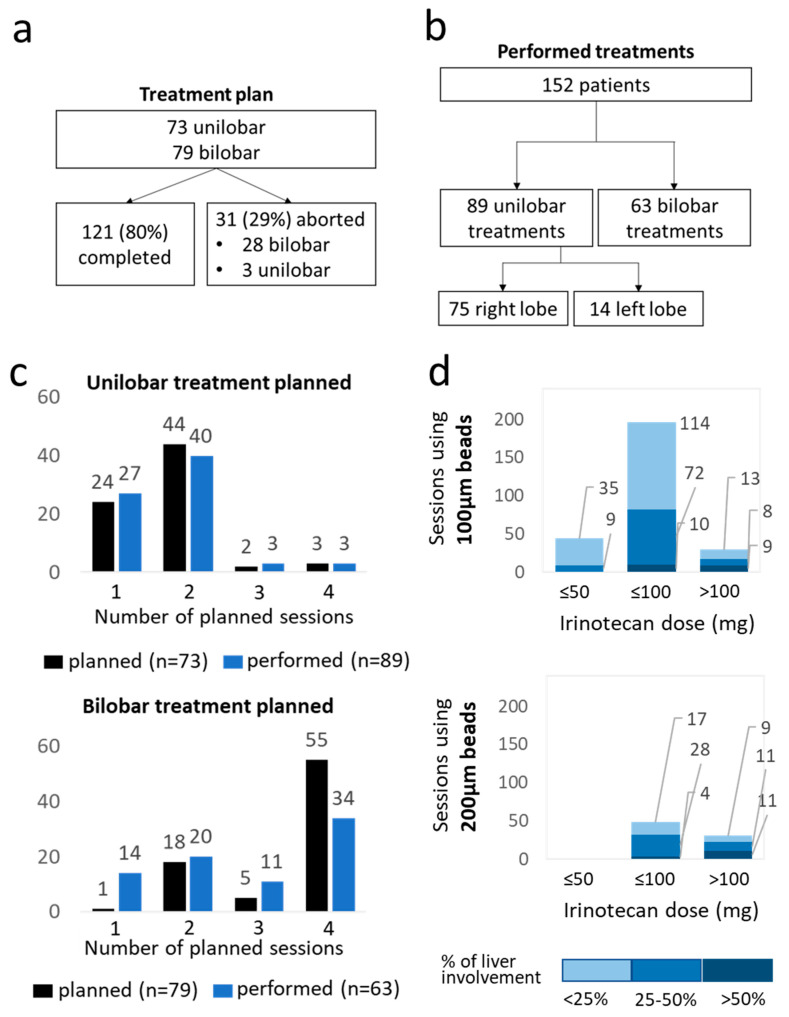
Technical considerations in terms of number of planned vs. performed treatment sessions and bead sizes and irinotecan doses used. Flowchart representing the number of planned uni- and bilobar tents, number of patients that did not complete the treatment plan (**a**). Flow chart representing the resulting number of performed uni- and bilobar treatments (**b**). The number of planned (dark blue) and actually performed (light blue) sessions per patients for uni- and bilobar treatment schedules (**c**). The number of sessions performed with 100 µm and 200 µm beads loaded with ≤50 mg, ≤100 mg and >100 mg irinotecan. The colours indicate the percentage of liver involvement from dark blue (>50%), medium blue (25–50%) and light blue (<25%) (**d**).

**Figure 2 jcm-11-06178-f002:**
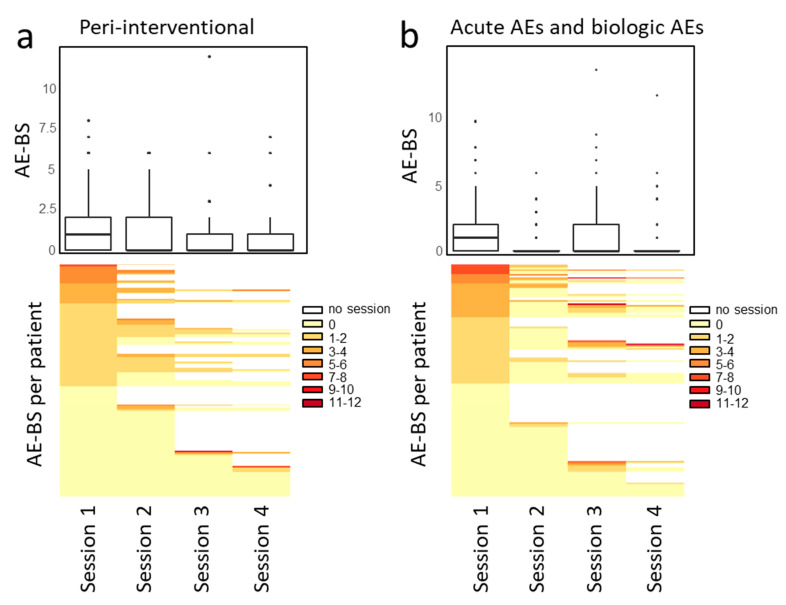
AE-BS per treatment session for peri-interventional (**a**) and acute and biologic AEs (**b**). The AE-BS represents the sum of all AE grades per patient. The boxplots show the AE-BS per treatment session for all patients and the heatmaps show the AE-BS per treatment session for each patient. Each row represents one patient. No clustering was used but the scores were sorted by descending AE-BS. White fields indicate no session was performed.

**Figure 3 jcm-11-06178-f003:**
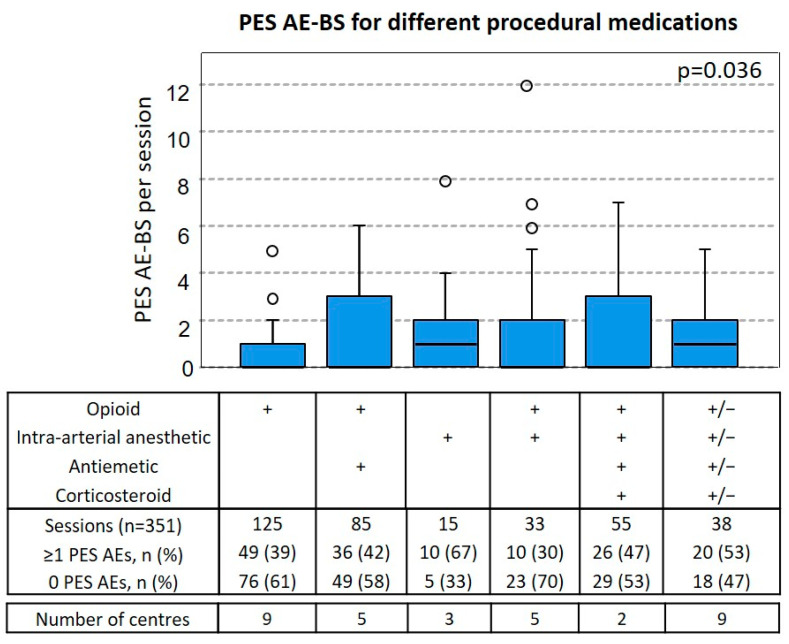
AE-BS of PES after the use different procedural medications. Boxplots of PES AE-BS as the sum of all PES AE grades per patient shown for different groups of procedural medications used per session. “+” indicates the use of the respective medication, “+/−” indicates the use for some but not all sessions in that group. The number of total sessions the number of sessions with at least ≥1 PES AE (percentage per group), as well as the number of sessions with at least 0 PES AE (percentage per group) and the number of different centres per group are listed. Statistically significant differences in the medians between all groups were assessed using the Kruskal-Wallis-Test.

**Table 1 jcm-11-06178-t001:** Baseline, primary tumour and liver metastases characteristics. Data is presented as counts and percentages.

Performance Status (ECOG) *n* (%)					
0				89 (59)	
1				52 (34)	
2				10 (7)	
3				1 (1)	
**Primary tumour location, *n* (%)**					
Rectum (from the anal verge to 15 cm above)				45 (30)	
Right colon (before splenic flexure)				35 (23)	
Left colon (after splenic flexure)				72 (47)	
**TNM status of the primary tumour, *n* (%)**					
Tis		N0	22 (14)	M0	40 (26)
T1	9 (6)	N1a	22 (14)	M1	87 (57)
T2	17 (12)	N1b	29 (19)	Mx	25 (16)
T3	98 (64)	N1c	13 (9)		
T4	28 (18)	N2a	26 (17)		
		N2b	21 (14)		
		Nx	19 (13)		
**Time since primary cancer diagnosis, *n* (%)**					
Synchronous (<6 months)				103 (68)	
Metachronous (>6 months)				49 (32)	
**Previous lines of systemic chemotherapy *n* (%)**					
no lines				24 (16)	
1–2 lines				112 (73)	
3 or more lines				16 (11)	
**Previous treatment with systemic irinotecan *n* (%)**				89 (58)	
**Previous ablation on liver metastases *n* (%)**				17 (11)	
**Previous intra-arterial treatment on liver metastases *n* (%)**				17 (11)	
**Location, *n* (%)**					
Whole liver				87 (57)	
Only left liver lobe				10 (7)	
Only right liver lobe				55 (36)	
**% of liver involvement, *n* (%)**					
<25%				82 (54)	
25–50%				59 (39)	
>50%				11 (7)	
**Number of lesions, *n* (%)**					
1				25 (16)	
2–3				46 (30)	
4–10				52 (34)	
>10				29 (19)	
**Extrahepatic metastases, *n* (%)**					
Yes				66 (43)	
No				86 (57)	
**CEA increased, *n* (%)**				80 (53)	
**CA 19.9 increased, *n* (%)**				64 (42)	
**Molecular characterisation**					
**RAS, *n* (%)**			**BRAF, *n* (%)**		
Yes	49 (32)		Yes	9 (6)	
No	56 (37)		No	60 (39)	
N/A	47 (31)		N/A	83 (55)	
**Treatment intention *n* (%)**					
First line treatment or consolidation therapy after response to first line				41 (27)	
Combination treatment with ablation with a curative intent				19 (13)	
Intensification of treatment with/without concomitant therapy				41 (27)	
Salvage treatment in progressive patients				46 (30)	
Other				5 (3)	
**Number of treatments per patients**					
1				41 (27)	
2				60 (39)	
3				14 (9)	
4				36 (24)	
5				1 (<1)	
**Bead size per treatment session (µm)**					
100				270 (77)	
200				80 (23)	
400				1 (<1)	
**Dose per treatment session (mg)**					
≤50				44 (13)	
≤100				246 (70)	
>100				61 (17)	
**Treatment plan**					
Completed				121 (80)	
Aborted				31 (20)	
**Technical success**					
Yes				346 (99)	
No				5 (1)	
**Technical success due to**					
Complete stasis				59 (17)	
Complete delivery of the dose				212 (60)	
Both				75 (21)	
**No technical success due to**					
No complete delivery of the dose due to				2 (<1)	
Immediate spasm of the artery				3 (<1)	

**Table 2 jcm-11-06178-t002:** AEs after LP-irinotecan-TACE treatment sessions graded according to CTCAE 4.3/5.0. Data is presented as counts and percentages.

**Adverse Events at the Same Day as a Treatment Session**				
Total AEs	269			
Patients with at least 1 AE, *n* (%)	91 (60)			
Total grade 3 or 4	19			
Patients with at least 1 grade 3 or 4 AE, *n* (%)	12 (8)			
	**Grade 1**	**Grade 2**	**Grade 3**	**Grade 4**
Infusion related reaction	0	0	1	0
Post-embolization syndrome	0	0	0	0
Pain	87	34	4	0
Nausea/vomiting	66	18	8	0
Fever	9	1	0	0
Hypertension	0	1	1	0
Diarrhea	0	1	0	0
Hemorrhage (liver subcapsular hematoma)	1	0	0	0
Hypertonia	0	0	3	0
Platelet count decreased	2	0	0	0
Other	3 ^1^	6 ^2^	2 ^3^	0
**Adverse events within 30 days after any treatment session**				
Total AEs	115			
Patients with at least 1 AE, *n* (%)	49 (32)			
Total grade 3 or 4	33			
Patients with at least 1 grade 3 or 4 AE, *n* (%)	21 (14)			
	**Grade 1**	**Grade 2**	**Grade 3**	**Grade 4**
Pain	15	18	4	0
Fever	8	2	0	1
Nausea/vomiting	6	10	5	0
Diarrhoea	2	2	0	0
Alopecia	0	0	1	0
Hepatic failure	0	0	1	0
Cholecystitis	0	1	1	0
Vertigo	2	0	0	0
Asthenia/Fatigue	0	2	0	0
Neutropenia	1	0	0	2
Increase in serum creatinine	1	0	0	0
Increase in alkaline phosphatase	28	4	0	0
Increase in ALT	21	3	2	0
Increase in AST	19	3	0	0
Blood bilirubin increase	6	0	2	1
Thrombocytopenia	2	0	0	0
Lymphocyte count decreased	23	4	0	0
Hypoalbuminemia	22	6	1	0
LDH	8	0	0	0
Other	4 ^4^	5 ^5^	6 ^6^	4 ^7^

^1^ Sinus bradycardia, vertigo, nontarget embolization of falciform artery with skin rash. ^2^ Anorexia, discomfort, fatigue, GOT GPT increased, lower limb edema, dislocation of one coil during coiling of Art. Cystica. ^3^ Abscess, cholecystitis. ^4^ Bradycardia, flatulence, obstipation, nontarget embolization of falciform artery with skin rash. ^5^ Hematoma groin, Lower limb oedema, Ascites, Anorexia, Abdominal Abscess. ^6^ Hepatic infection, osteolysis, infection, CRP increasing, renal failure + hyperkalemia, abscess. ^7^ Sepsis, colonic obstruction, stomach perforation.

**Table 3 jcm-11-06178-t003:** Multivariable binary logistic regression analysing the relationship between baseline- and treatment associate factors and sessions with ≥1 peri-interventional AEs. Following variables were considered in the multivariable model: ECOG, treated liver lobes, percentage of liver involvement, bead size, dose, session number and total number of sessions performed.

Variable	Reference/Units of Increase	Odds Ratio (95% CI)	*p* Value
Treated liver lobes	unilobar	1	
	bilobar	3.1 (1.5–6.4)	0.002
Percentage of liver involvement	<25%	1	
	>25%	2.9 (1.8–4.9)	<0.001
Bead size	100 µm	1	
	>100 µm	3.9 (2.0–7.5)	<0.001
Dose	1 mg	0.99 (0.98–1.0)	<0.001
Session number	1 session	0.7 (0.5–0.9)	0.006
Total number of sessions performed	1 session	0.6 (0.4–0.8)	0.003

**Table 4 jcm-11-06178-t004:** Multivariable binary logistic regression analysing the relationship between baseline- and treatment associate factors and sessions with ≥1 acute AEs. Following variables were considered in the multivariable model: ECOG, treated liver lobes, number of lines of liver-directed systemic therapy, dose and total number of sessions performed.

Variable	Reference/Units of Increase	Odds Ratio (95% CI)	*p* Value
Treated liver lobes	unilobar	1	
	bilobar	3.2 (1.4–7.5)	0.06
ECOG	0	1	
	>0	3.3 (1.8–6.1)	<0.001
Dose	1 mg	0.99 (0.98–1.00)	0.04
Total number of sessions performed	1 session	0.5 (0.3–0.7)	<0.001

**Table 5 jcm-11-06178-t005:** PES AEs per session in absence of corticosteroid use compared to sessions with corticosteroid use. Data is presented as counts and percentages. Significant differences between categorical data were assessed using Chi-squared test (*p* values ≤ 0.05 were considered significant).

PES AE per Session	Grade	No Corticosteroids, *n* (%) *n* = 277	Corticosteroids + Antiemetic, *n* (%) *n* = 74	*p* Value
Fever				
	G1	9 (3)	0 (0)	ns
	G2	1 (<1)	0 (0)	ns
Nausea				
	G1	32 (12)	5 (7)	ns
	G2	9 (3)	5 (7)	ns
Vomiting				
	G1	36 (13)	2 (3)	0.01
	G2	6 (2)	2 (3)	ns
	G3	8 (3)	0 (0)	ns
Pain				
	G1	76 (27)	10 (14)	0.01
	G2	22 (8)	14 (19)	0.009
	G3	3 (1)	0	ns

ns = not significant (i.e., *p* value >0.05).

## Data Availability

Not applicable.

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
