# Peer review of "Safety, Feasibility and Technical Considerations from a Prospective, Observational Study—CIREL: Irinotecan-TACE for CRLM in 152 Patients"

_jcm, 2022, doi:10.3390/jcm11206178_

Round 1

Reviewer 1 Report

1. You report that 60% of patients experienced at least 1 peri-interventional AE of any grade. This appears high compared to prior studies (e.g., Gruber-Rouh et al etc) who reported only 10%. Please cite other relevant studies and compare your findings to those of other studies. 

2. The most clinically relevant part is the establishment of risk factors for AEs. You need to perform a logistic regression analysis to find the independent predictors of AEs.

3. You mention that "Data was analysed and plotted using RStudio 165 under R4.0.0 or IBM SPSS Statistics 25." Which part of the analysis was conducted by using SPSS package?

Reviewer 2 Report

This observational international multicentric study, provides important informations about irinotecan TACE treatment.

The paper reports safety and feasibility data. the most important data concern the management of post embolization syndrome and in particular the management of pain.

minor fixes: reference 1 doesn't seem to fit

Many references cited, could the authors reduce the number of references to facilitate reading

Round 2

Reviewer 1 Report

The authors addressed my concerns.